# Research on the Symbolic 3D Route Scene Expression Method Based on the Importance of Objects

**Fulin Han** [1] , **Liang Huo** [1,2], **Tao Shen** [1,2,*], **Xiaoyong Zhang** [3,*], **Tianjia Zhang** [1] **and Na Ma** [1]

1. School of Geomatics and Urban Spatial Information, Beijing University of Civil Engineering and Architecture, Beijing 100044, China
2. Key Laboratory of Urban Spatial Information, Ministry of Natural Resources, Beijing 100044, China
3. Chinese Academy of Fishery Sciences, Beijing 100141, China
* Correspondence: shentao@bucea.edu.cn (T.S.); zhangxiaoyong@cafs.ac.cn (X.Z.)

**Abstract:** In the study of 3D route scene construction, the expression of key targets needs to be highlighted. This is because compared with the 3D model, the abstract 3D symbols can reflect the number and spatial distribution characteristics of entities more intuitively. Therefore, this research proposes a symbolic 3D route scene representation method based on the importance of the object. The method takes the object importance evaluation model as the theoretical basis, calculates the spatial importance of the same type of objects according to the spatial characteristics of the geographical objects in the 3D route scene, and constructs the object importance evaluation model by combining semantic factors. The 3D symbols are then designed in a hierarchical manner on the basis of the results of the object importance evaluation and the CityGML standard. Finally, the LOD0-LOD4 symbolic 3D railway scene was constructed on the basis of a railroad data to realise the multi-scale expression of symbolic 3D route scene. Compared with the conventional loading method, the real-time frame rate of the scene was improved by 20 fps and was more stable. The scene loading speed was also improved by 5–10 s. The results show that the method can effectively improve the efficiency of the 3D route scene construction and the prominent expression effect of the key objects in the 3D route scene.

**Keywords:** three-dimensional route scenes; object importance; calculation of spatial importance; three-dimensional symbolic hierarchical design; symbolic representation

## 1. Introduction

In recent years, digital maps are gradually evolving from two-dimensional planes to three-dimensional scenes due to the progress of the web, multimedia, three-dimensionality, and multi-temporality in the direction of the new era. Maps are the result of human spatial cognition, and they are also a basic tool for spatial cognition [1]. Compared with traditional two-dimensional maps, three-dimensional scenes have obvious superiority in geospatial performance, and users can more accurately grasp the relevant information of various geographical entities [2]. This is particularly the case for road, railway, and other route scenarios. The data are characterised by a ribbon distribution, a complex data structure, and a large number of data sources. Therefore, an efficient 3D scene representation is of great importance for the user's browsing efficiency [3].

Many scholars have conducted a large amount of research on this issue, and existing research on fast visualisation schemes for 3D scenes has been more productive.

Chaturverdi et al. [4] developed an interactive web client based on semantic 3D city models using HTML5 and WebGL. Wendel et al. [5] developed a plugin-free web visualisation of semantic 3D city models in Karlsruhe, Germany. Farkas [6] compared the most capable open-source web GIS libraries on the basis of their feature coverage. Chen et al. [7] developed a workflow for visualising BIM (Building Information Modelling) models on CesiumJS. Resch et al. [8] developed a web-based 3D interface for visualising

marine data and pointed out the problem of visualising four-dimensional (4D) data (time as the fourth dimension). Tang Shengjun et al. [9] dissected the geometric, topological, and combinatorial relationships of the facility structure and constrained the scene objects through parametric modelling to achieve linkage between scene and parameters and dynamic update of the scene. Wang Wei et al. [10] designed a feature-point-based terrain adaptation method for the terrain and scene object separation modelling method to achieve multi-scale fusion of terrain and features. Putri et al. [11] used 3D GIS to map the location of drinking fountains and model the distribution of pipe networks to plan and monitor drinking water supply systems while assessing whether the water supply plan met the water demand. Liu et al. [12] used an online walkthrough for large underground scenarios to achieve low resource consumption and high efficiency in a large-scale scenario web by lightening the raw data, reconstructing the scenario structure, and integrating the scenario management. Zhou et al. [13] proposed a lightweight framework for S-LCM compression-driven Web3D for bandwidth and browser load limitations, accomplishing the task of displaying 3D models on a web browser through mesh compression and segmentation.

The visual representation of lines has also been studied and explored by many scholars. In terms of platform software, some of the more famous ones are IB&T's CARD/1 software, which is powerful in 3D modelling and visualisation and can perform many tasks such as road planning and design [14]; Bentley's Power InRoads [15] and MXROAD, both of which are auxiliary design software for designing transport infrastructure designs, have 3D drawing, and have Freeworld3D that provides interactive terrain editing, which can raise, lower, and level the terrain according to user needs; and Unity 3D scene editor, which supports real-time input, feedback, and updates in 3D scenes [16,17], and terrain reconstruction can be performed by creating new models; in addition, GIS platform software such as ArcGIS [18] and SuperMap are also available. SuperMap and other GIS platform software are also applied to visualise and model 3D railway scenes due to their powerful 3D functions. GERail, jointly developed by the Second Institute of China Railway and Central South University, makes up for the difficulties in selecting lines in areas where geographic information is scarce by invoking publicly released terrain data, providing an intuitive and convenient way to compare line options and report presentations [19]. The virtual high-speed railway scene modelling service system developed by Southwest Jiaotong University organises and manages the object models of high-speed railway scenes with complex spatial relationships and a wide variety of scenes, realising rapid modelling of 3D high-speed railway scenes [20]. The pavement inspection and assessment system and road maintenance management platform of China Kaidawan [21] use the advantages of GIS and road BIM [22] information integration and visualisation to realise the macro management of the whole line of GIS and the fine management of BIM, so that the basic data of the road can be transferred effectively. It provides a simulated real-world reference for road maintenance to formulate disposal plans and scientific decision making and can visually display the road operation status.

In terms of technical research, Han Yuan [23] proposed a road automatic selection model based on AHP by constructing road contextual feature indicators to achieve a more reasonable road selection after integrating multiple attributes; Li Xiaoxuan [24] focused on the theory and technology related to geographic information representation for cartographic synthesis, and proposed a model for building road attribute information and data structure model for cartographic synthesis, which provides a new method for road scene; Jiang Hongfei et al. [25] proposed a method to generate a local triangular network by using the intersection of railway roadbed profile lines and DEM grid lines, which realised the establishment of a 3D terrain and overall model of the line; and Zhang Yanhui [26] designed an algorithm for generating railway roadbeds and slopes to achieve effective integration of line models and terrain models in 3D scenes. Shi Shaohua [27] used SketchUp for scene modelling and ArcGIS-Engine-based programming to develop a visualisation system, providing an important reference for the construction of virtual scenes along railway lines. Pu Hao et al. [28,29] proposed a method to construct and simplify the 3D overall model of

the road while taking into account constraints, which provides a reference for the network transmission of massive scene data. Wang Jinhong and Zhu Jun et al. [30] achieved the positioning and semantic addition of scene objects by establishing a linear reference model to construct 3D railway scenes. Ma Xiaolong et al. [31] proposed segmenting railway scene objects and making 3D symbols, and then storing 3D symbols in the form of a database to achieve symbolic modelling.

There is still the problem of how to appropriately simplify the complex model in the scene during the construction of the 3D line scene. That is, how to choose the right algorithm to simplify the drawing of the model in the scene while ensuring the 3D visual effect. This led to the emergence of LOD (Level of Details) technology [32]. According to the principles of human vision, objects that are close to the human eye are clearer and have higher accuracy, while on the other hand they are rougher, i.e., less accurate [33]. In 1996, Lindstrom [34] implemented DEM triangulation, using the quadtree principle to construct a continuous LOD model, which saved mapping time to some extent, but was computationally intensive and had an impact on the rendering efficiency [35]. Hoppe [36] has since designed a Progress Mesh model and combined it with the LOD model to better solve the polygon mesh problem, but this model causes a large memory drain, complicating the display of large amounts of data and increasing the strain on the GPU. In order to save time in terrain simplification, an algorithm using ROAM to pre-process the LOD emerged, and the Chunked LOD model [37] was constructed to alleviate the pressure on the GPU, but the model took up a large amount of hard disk space. According to the characteristics of 3D model data, Chen Jing et al. [38] constructed a 3D model data structure based on point indexing that uses a multi-level LOD management model for image textures, thus constructing a 3D model with LOD structure. Tan Qingsheng et al. [39] used an efficient compression of 3D scenes as well as scheduling and rendering algorithms to enhance the rendering performance of the scenes. Although this method alleviates many of the problems associated with data visualisation due to network transmission, its data format model is not uniform, thus making interactive sharing between users impossible.

Although the above-mentioned studies have solved the problem of constructing 3D line scenes to a certain extent, they have mainly focused on the study of large-scale urban 3D scene representation or the study of 3D tile data organisation and loading patterns [40]. These studies lack the design of representation rules for 3D route scenes dominated by the spatial cognitive needs of users and the importance evaluation models applied to 3D symbolic representations [41–44]. Thus, there are problems with the selection of key targets and the prominent expression of objects in the study of symbolic 3D route scenes. These limit the promotion and application of 3D maps.

To address these issues, this research proposes a method for symbolic 3D route scenes that takes into account the importance of physical objects. First, the spatial importance of similar elements in the 3D route scene needs to be calculated, and then these elements are processed according to semantic factors in order to create a corresponding importance evaluation model. This gives us the importance distribution in the study area. The 3D symbols are then classified and modelled using a symbolic approach based on the importance of each entity in the 3D route scene. Finally, all entities are displayed hierarchically at different scales, depending on their importance.

The research proposes a new form of 3D route scene representation that is based on user perception and highlights important objects. The method uses 3D symbols instead of real 3D models to represent 3D route scenes. Not only can it effectively emphasise the key objects in the scene, but it also makes the loading of the scene more stable and efficient. In addition, the construction method of the object importance evaluation model may also be useful for research in other application areas. In summary, the research proposes a new idea for the representation of 3D route scenes.

## 2. Materials and Methods

This research proposes a symbolic 3D route scene representation method that considers the importance of objects. Geographical entity objects need to be extracted from the 3D route scenes first. Using the constructed object importance evaluation model, as well as the attributes and spatial information of the geographical entity objects, the spatial importance and the semantic importance are calculated, respectively. Finally, the object importance is obtained. According to the ranking result, the geographic entity objects are stratified and used to construct symbolic 3D route scenes. The technical flow chart is shown in Figure 1.

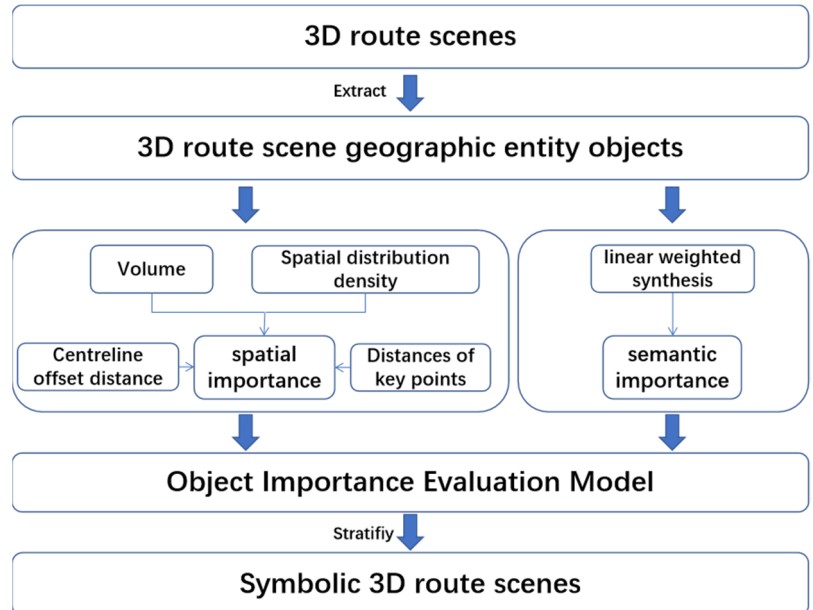

**Figure 1.** A technological roadmap for the construction of symbolic 3D route scenes according to the importance of objects.

### 2.1. Object Importance Evaluation Model

2.1.1. Grounded Theory of Object Importance Evaluation Models

In 3D route scenes, geographic entity objects are characterised by a variety of types, complex spatial relationships, and a banded distribution. Moreover, the 3D route scenes tend to focus more on the overall representation of the scene and pay less attention to the features of individual models. Therefore, the scene can be represented by selecting important geographic entity objects and using a simplified 3D symbolic notation. This method enhances the outstanding representation of the key objects.

To respond to the way in which to select the important geographical entity objects in the three-dimensional route scene expression, the importance of the geographic entity objects in the 3D route scene needs to be judged so that the more important geographical entity objects can be expressed in priority.

When constructing a 3D scene, the research related to 3D building synthesis can be referenced. The main objective of 2D map synthesis is that not only can map users clearly see the map image, but also that information on the specific characteristics and spatial location of geographical elements can be presented. The 3D scene can be seen as an extension of the 2D map in 3D space. The user of a 3D scene can obtain the geometric and textural features of a 3D geographic entity in the most realistic way. At the same time, it contains the spatially distributed morphological features of the 3D model in its entirety. Three-dimensional scenes should represent the spatial characteristics of geographic entity objects more graphically to the viewer. On the other hand, 3D models transmitted on the basis of the Internet have a huge amount of data and must deal with the issue of expression priority in the expression process [45].

2.1.2. Spatial Importance Calculation of Objects in 3D Route Scenes

The importance level is the basis for the selection of objects for the 3D route scenes. It is based on certain indicators that classify entities into different levels and orders them from highest to lowest. The importance rating of an entity object must take into account a variety of indicators such as type, class, density, and spatial distribution characteristics in order to ensure that the evaluation is as comprehensive and scientific as possible. In practice, it is not easy to obtain a satisfactory selection result if only one specific indicator is used. Therefore, several indicators are usually used for comprehensive assessment and combined implementation.

Semantic and spatial information can be expressed directly in the 3D scene by visualisation. The semantic information includes administrative levels, building materials, population numbers, etc. It is highly subjective and generally not computable or comparable. Therefore, the method of mapping functions must be used to make it into a numerical type that can be mathematically calculated and compared. The spatial information includes location, shape, size, distribution density, spatial structure, etc. A comprehensive calculation of multiple spatial information indicators can effectively compare the importance levels of similar objects in terms of spatial attributes [46].

In this research, four importance level indicators were considered to calculate the spatial importance of objects, namely, centreline offset distance, volume, spatial distribution density, and distances of key points. These provide an objective basis for the construction of object importance evaluation models in 3D route scenes.

(1)    Centreline offset distance

Distance from the centre point of the object's base to the projection point of the road centreline. The smaller the result, the more important it is.

(2)    Volume

Volume is the most intuitive indicator of an object. As a rule, the larger volume is more important than the smaller one. The formula is given in Equation (1).

$$f_{size} = \frac{V}{\frac{1}{n} \times \sum_{i=1}^{n} V_i} \tag{1}$$

(3)    Spatial distribution density

The totality of a 3D route scene is generally greater than the individual characteristics of the objects. Spatially distributed objects are therefore more representative of the overall situation of a 3D route scene and are more important than densely distributed objects.

The average step length T is calculated according to the formula T = L/C, where T is the average step length, L is the length of the road segment where all objects of that type are located, and C is the total number of objects of that type. The density of the spatial distribution can be obtained from the size of the average step size T. As the average step length increases, the spatial distribution widens while the density decreases.

(4)    Distances of key points (junctions and corners)

In a 3D route scene, there are key points on the route that can be used as critical factors in the calculation of spatial importance. Road junctions and corners are key spatial feature locations in the construction of 3D scenes, as often the geographic entity objects at road junctions and corners are more important. Road junctions are uncertain due to the number of roads at the intersection and the type of road. Therefore, it is first necessary to determine the importance level of road intersections according to different situations and then calculate the spatial importance of objects according to their distance from the intersection [47].

In order to standardise the importance rating of road junctions, a crossing importance target (CIT) is used. The CIT has a clear and prominent calculability and reasonableness for all types of junction situations. The methodology is as follows:

1. If no other road crosses or borders this junction, then $CIT = 1 \times CI$ for this junction.
2. If the end point of a road of importance CI joins this junction, then the junction weight w for this road counts as 2, i.e., $CIT = CIT + 2 \times CI$.
3. If a road of importance $CI$ crosses this junction, the junction weight w counts as 3.5, i.e., $CIT = CIT + 3.5 \times CI$.

Therefore, assuming that there are $N$ roads crossing or intersecting the junction, the importance of the junction is shown in Equation (2). And examples are given to illustrate the results of the calculation for four actual road conditions, as shown in Figure 2.

$$CIT_I = \sum_{j=0}^{N} w_j \times CI_J \tag{2}$$

where $w_j$ is the road junction weight, and $CI_J$ is the importance of the road, e.g., class, speed, material.

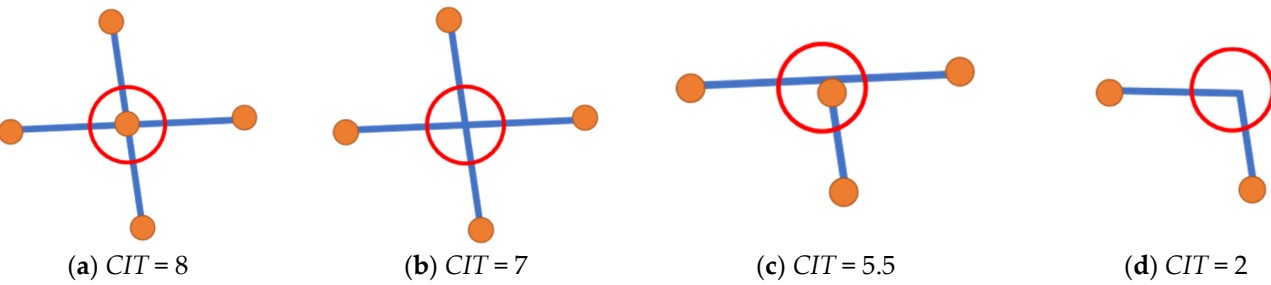

(**a**) $CIT = 8$  (**b**) $CIT = 7$  (**c**) $CIT = 5.5$  (**d**) $CIT = 2$

**Figure 2.** Road junction and corner importance target: Assuming that the importance of each road $CI$ is 1, the $CIT$ quantitative calculations are as follows. (**a**) Four roads intersect at a point, and each road is spatially connected to the junction, so the junction $CIT = 2 + 2 + 2 + 2 = 8$. (**b**) Two roads intersect at a single point, and each road crosses the junction in spatial relationship; therefore, the junction $CIT = 3.5 + 3.5 = 7$. (**c**) Two roads intersect at a point, one road crosses the junction, and the other road joins the junction, so the junction $CIT = 3.5 + 2 = 5.5$. (**d**) A corner of the road itself, so that the corner $CIT = 2$.

Combining the above four importance level indicators, the formula for calculating the spatial importance of objects in 3D route scenes is shown in Equation (3).

$$f_{spatial} = \begin{cases} \dfrac{\sum_{j=1}^{m} CIT_j \times \left[\frac{(r-d)}{r}\varnothing + W_j \frac{(r'-d')}{r'}\varnothing'\right] + \frac{V}{\frac{1}{n} \times \sum_{i=1}^{n} V_i}}{C} \times L, & d \leq r \\ 0, & d > r \end{cases} \tag{3}$$

where $m$ is the number of roads; $CIT$ is the importance index of the current road; $W_j$ is the importance weight of the junction and the corner; $r, r'$ are the buffer radii of the road and the key point, respectively; $d, d'$ are the distances from the object to the road and the key point, respectively; $\varnothing, \varnothing'$ are the user-definable coefficients of road relevance; $V$ is the current 3D model volume; $n$ is the number of 3D models; $V_i$ is the volume of each 3D model; $L$ is the length of the road segment where all objects of this type are located; and $C$ is the number of all objects of this type.

### 2.1.3. Object Importance Evaluation Model Construction

In practice, in addition to the spatial importance of the object, the influence of semantic information factors needs to be considered. The semantic importance of a geographic entity object is determined by a variety of factors, such as geographic element codes, construction materials, and administrative hierarchy. These factors are needed to be considered comprehensively for obtaining the semantic importance of an object. The

weight calculation model can provide an effective solution for the multi-factor synthesis of geographic entity objects.

The weighting approach, which takes into account the relevance of the indicators, takes into account both the influence of the degree of variation of each attribute across the evaluation objects on the weights and the influence of the correlation between the individual attributes. This method uses more comprehensive and complete information and allows the impact of each attribute to be weighed in its entirety. It is therefore a more desirable method for determining weights [48].

With the weights determined, a semantic importance evaluation model can be built through the calculation method of linear weighted synthesis. The semantic importance evaluation model is shown in Equation (4).

$$f_{semantics} = \sum_{i=1}^{n} \omega_i X_i \tag{4}$$

where $\omega_i$ is the weight of the attribute; $X_i$ is the value of the attribute; and $n$ is the number of attributes.

In summary, the object importance evaluation model in this research needs to take into account the calculation results of both the spatial importance and the semantic importance of the geographical entity objects of the 3D route scene. The calculation formula is given in Equation (5).

$$f_{importance} = f_{spatial} * f_{semantics} \tag{5}$$

### 2.2. Importance-Based Hierarchical Classification and Modelling of 3D Route Scene Symbols

The real world has a wide variety of things and complex structures. This poses a number of difficulties for the construction of 3D geographical scenes. Therefore, the classification of 3D symbols need to be achieved through the classification of geographical entity objects. This allows for the simplification of large, complex, and difficult-to-handle data information [49].

The main object of study is 3D route scene geographic element symbols. On the basis of the natural classification, the object importance evaluation model is introduced to further deepen the stratification [50]. The flow chart of the experiment is shown in Figure 3.

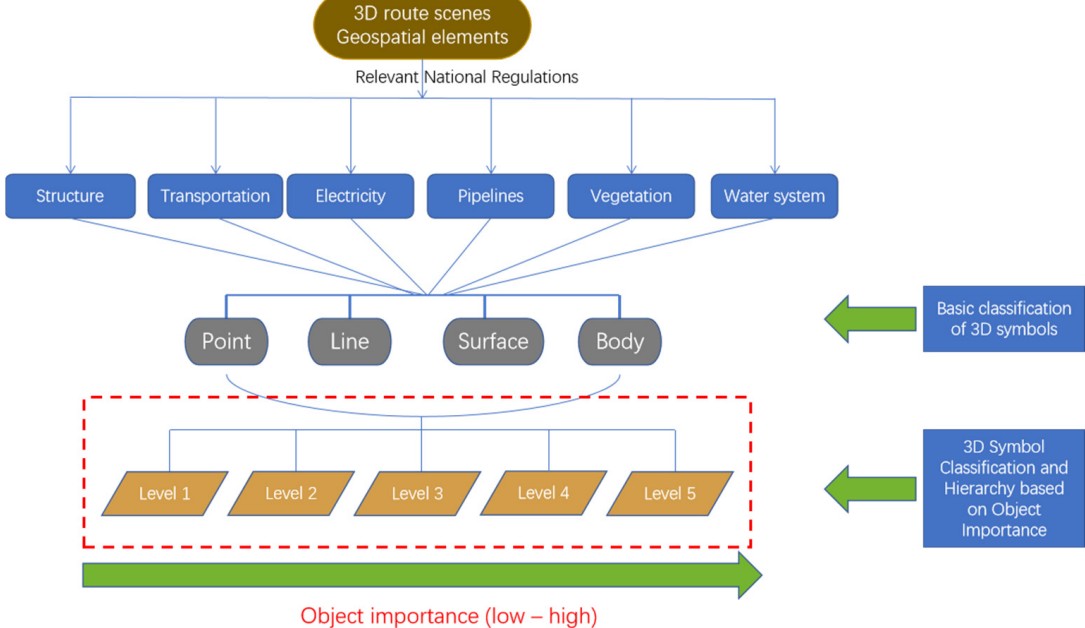

**Figure 3.** Flowchart: 3D symbol classification and hierarchy.

In order to standardise the research and reduce ambiguity, the specific research content took the relevant standard specifications of the national basic scale map as the main reference material, which mainly includes the National Basic Scale Map Format, the National Basic Scale Map Compilation Specification, the Basic Geographic Information Elements Data Dictionary, and the 3D Geographic Information Model Data Product Specification [51–53]. According to the above relevant specifications, the geographical elements of 3D route scenes can generally be divided into structure, traffic, electricity, pipelines, vegetation, and the water system. As shown in the Figure 4, the different 3D route scene geographic elements were firstly divided into the above six elements.

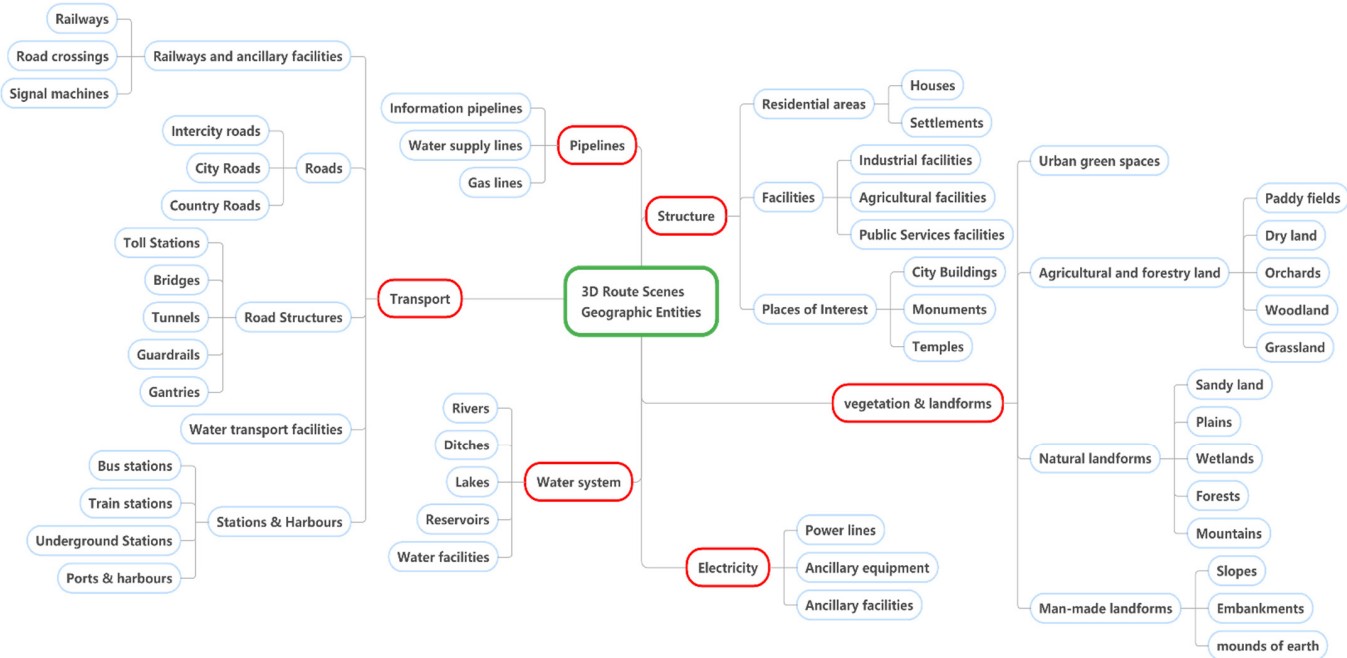

**Figure 4.** Natural classification of geographic entities for 3D route scenes. It has combined specific route scenes in reality according to relevant specifications. A classification of the geographical entities that may appear in the construction of 3D route scenes was made.

An initial natural classification of all objects was performed. Subsequently, on the basis of the initial natural classification, all objects in the 3D route scene were classified into five levels according to their importance. In this way, a secondary classification hierarchy of elemental objects was achieved. Finally, a 5-level map notation for the 3D route scene was designed by combining the ideas of the 5-level LOD (Levels of Detail) model of CtiyGML, as shown in Figure 5.

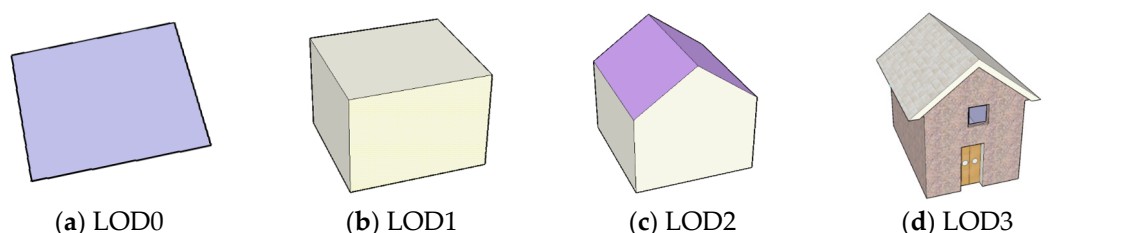

(**a**) LOD0     (**b**) LOD1     (**c**) LOD2     (**d**) LOD3     (**e**) LOD4

**Figure 5.** The five LODs of CityGML: (**a**) LOD0: 2D symbol; (**b**) LOD1: simple geometry; (**c**) LOD2: simple geometry combinations; (**d**) LOD3: geometry with realistic textures; (**e**) LOD4: real internal structure.

Three-dimensional map symbols differ from the abstract characteristics of 2D map symbols. They not only reflect the essential features and general properties of a feature,

but more importantly, also represent the complex surface property information of a feature. However, due to the limitations of computer processing power and cost. It is often difficult and inefficient for us to reconstruct all the detailed details of a feature. Therefore, symbol simplification becomes one of the important processes in the design of 3D map symbols. Symbol simplification includes the simplification of geometric details and the simplification of textural details [54]. The purpose of these are to remove unimportant details and retain the parts that best represent the distinguishing features of the feature, thus meeting the need for efficient and realistic visualisation. The 3D map notation is a simplified reflection of the objective, in order to discover and understand the essential properties and basic laws of the objective. Therefore, it makes sense to exaggerate the important features of the 3D map symbols and ignore the secondary features according to the actual situation.

The modelling of 3D route scene map symbols needs to be designed according to the semantic information and geometric spatial structure of the geographic entity objects, as well as satisfying the multi-detail hierarchy. For low-level-of-detail symbols, the abstraction of the symbols is fully utilised to show less detail of the objects and to minimise the loss of system performance from a large range of models. For high-level-of-detail symbols, the accuracy and realism of the symbols can be improved while ensuring system performance [1]. The importance of the geographical entity objects in the 3D route scene needs to be ranked, the higher the importance level the more levels of detail in the designed 3D symbols.

*2.3. Multi-Scale Representation of Symbolic 3D Railway Scenes*

The symbolic 3D route scene is a form of representation that highlights the objects of focus for the user on the basis of the spatial perception of the geographical object [2]. Through the object importance evaluation model established above, it is possible to analyse the regions and geographical entity objects that users focus on. Combined with the classification results of 3D symbols obtained previously, this research defines five levels of symbols from low to high on the basis of object importance. The symbolic 3D route scenes can be expressed for the geographic entity objects in the scenes in order according to the symbolic levels.

In this research, a multi-scale representation method for symbolic 3D railway scenes was designed on the basis of object importance. The method combines the object importance of geographical entities in the scene and the 5-layer LOD idea of CityGML. The symbolic 3D railway scene was divided into a 5-layer scale, including a 5-layer 3D railway scene detail hierarchy model from LOD0-LOD4, as shown in Table 1. When users browse the 3D scene, they can observe the symbolic 3D railway scene at different levels by changing the viewing distance. The final result is a multi-scale representation of the symbolic 3D railway scene.

**Table 1.** Three-dimensional railway scene detail hierarchy model.

| Levels of Detail | 3D Symbol Construction Standards | Description |
| --- | --- | --- |
| LOD0 | The technical means of 3D visualisation ensures that 2D symbols are always in front of the screen from different viewpoints, so that they are correctly represented. | Contains Level 5 objects. |
| LOD1 | It consists of simple geometry without complex geometric transformations, Boolean operations, etc., and this level model is not given a texture material. | Contains Level 4 and Level 5 objects. |
| LOD2 | The hierarchical model has a clear geometric outline, constructed by performing certain geometric transformations and Boolean operations on simple geometry. | Contains Level 3, Level 4, and Level 5 objects. |
| LOD3 | Constructed from simple geometry with complex geometric transformations and Boolean operations to produce realistic textures. | Contains Level 2, Level 3, Level 4, and Level 5 objects. |
| LOD4 | The use of fine-grained 3D models for maximum reproduction of geographic solid objects. | Contains Level 1, Level 2, Level 3, Level 4, and Level 5 objects. |

### 2.4. Experimental Data

The experimental data were derived from a high-speed railway line, containing relevant vector data, 3D model data, and business data. The experimental area is shown in Figure 6.

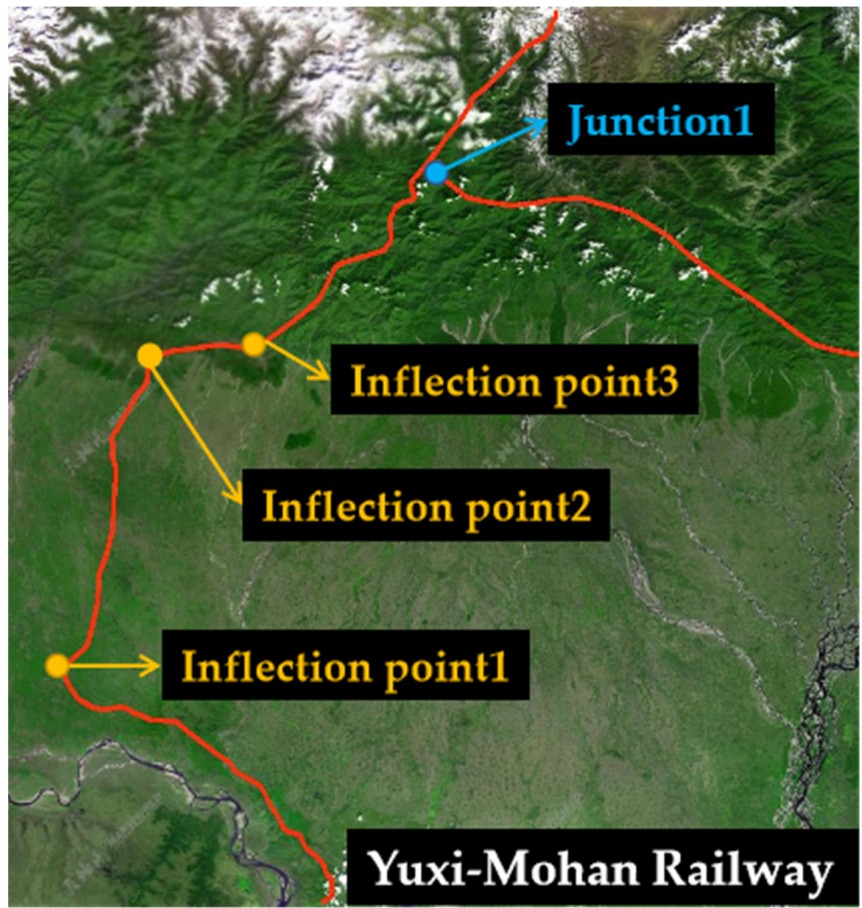

**Figure 6.** Range of experimental data. It contains a junction and three corners in two roads.

## 3. Results and Discussion

### 3.1. Object Importance Calculation for 3D Railway Scenes

Firstly, the locations and weights of key points in the selected road sections were determined. The results are shown in the Table 2.

**Table 2.** The locations and weights of key points.

| Location | Mileage | *CIT* |
| --- | --- | --- |
| Junction 1 | 89,822 | 2 + 3.5 = 5.5 |
| Inflection point 3 | 199,024 | 2 |
| Inflection point 2 | 241,022 | 2 |
| Inflection point 1 | 386,582 | 2 |

The four importance level indicators of centreline offset distance, volume, spatial distribution density, and distances of key points were determined on the basis of the data for the five types of elements: stations, tunnels, bridges, roadbeds, and signalling machines. The data parameters are shown in the Table 3.

**Table 3.** Five types of object data parameters.

| Category | CI (Road Importance Indicators) | $r'$ (Junction and Turning Point Buffer Radius) | L (Length of Road) | C (Number of Objects) |
|---|---|---|---|---|
| Stations | 1 | 20,000 | 554,260 | 19 |
| Tunnels | 1 | 20,000 | 554,260 | 91 |
| Bridges | 1 | 20,000 | 554,260 | 185 |
| Roadbeds | 1 | 20,000 | 554,260 | 247 |
| Signalling | 1 | 20,000 | 554,260 | 905 |

The spatial importance of all geographic elements in this category was calculated separately and combined with the semantic importance; the object importance was obtained, ranked, and divided into five levels according to the classification and proportion from highest to lowest, and the results are shown in Table 4.

**Table 4.** Object importance ranking results.

| Level | Geographical Elements | Object Importance (Rounded) |
|---|---|---|
| 5 | Yuanjiang Station | 160 |
| | Ning'er Station | 145 |
| | Guangyao Station | 145 |
| | Xishuangbanna Station | 145 |
| | Maohan Station | 140 |
| | Yanhe Station | 140 |
| | Ganzhuang Tunnel | 120 |
| | Huanian Station | 100 |
| | Mojiang Station | 100 |
| | … … | … … |
| 4 | Yueya Tian Tunnel | 80 |
| | Shitouzhai Tunnel | 71 |
| | Xinhua Tunnel | 71 |
| | Heping Tunnel | 71 |
| | Nam Lian Shan Tunnel | 71 |
| | Yau Yee Tunnel | 68 |
| | Hele Tunnel | 68 |
| | Tarko River Two-Lane Middle Bridge | 60 |
| | Ega Mountain Tunnel | 55 |
| | … … | … … |
| 3 | Four-Lane Middle Bridge over the Puma River | 30 |
| | Fo Tai Shan Special Bridge | 29 |
| | Xishuangbanna Twin-Lane Special Bridge | 29 |
| | Shanggang No.5 Special Bridge | 22 |
| | Xuejiashan Double-Lane Special Bridge | 21 |
| | Section 4 roadbed | 16 |
| | Yuanjiang Twin-Lane Special Bridge | 15 |
| | … … | … … |

**Table 4.** *Cont.*

| Level | Geographical Elements | Object Importance (Rounded) |
|:---:|:---:|:---:|
| | Section 9 roadbed | 10 |
| | Section 11 roadbed | 10 |
| 2 | Section 17 roadbed | 9 |
| | Section 1 roadbed | 9 |
| | Section 22 roadbed | 9 |
| | Section 3 roadbed | 8 |
| | . . . . . . | . . . . . . |
| | Huanian to Yuanjiang Section 1603 Signals | 4 |
| | Signal Tower 1347 | 3 |
| | Signal machine 3051 between Guangyao and Ning'er | 3 |
| 1 | Xishuangbanna 4392 signalling machine | 3 |
| | Yuxi S8 signalling machine | 3 |
| | Maohan S-issuing code point signal machine | 3 |
| | . . . . . . | . . . . . . |

*3.2. Symbolised 3D Railway Scene Construction*

On the basis of the results of the object importance evaluation model, we classified and stratified the geographical elements of the 3D route scenes in the experimental area. Then, we modelled the 3D symbols. The current mainstream 3D symbol modelling tools are ArcGIS, SketchUp, AutoCAD, 3D Studio MAX, CityEngine, etc. The main process of 3D symbol modelling is divided into the following steps:

① The design of the three-dimensional symbol model. In the process of modelling three-dimensional symbols, it is necessary to consider both the artistic nature of the symbols and also the intuitive expressiveness of the symbols. As the expression environment of three-dimensional symbols and two-dimensional symbols is different, three-dimensional symbol modelling should take into account four modelling principles: simplicity, abstraction and image balance, coordination, and feasibility.

② Modelling of three-dimensional symbolic models. According to the semantic information of the geographical entity objects and the two-dimensional topological spatial structure, we modelled the objects within the experimental scope separately. On the basis of the idea of symbolisation, we generated 3D models with a high degree of realism through the software for the models of objects with a high level of hierarchy in the region. For other object models with lower levels in the region, we were able to use a combination of simple three-dimensional geometries such as spheres, columns, and vertebrae to express them. Figure 7 shows the 3D symbol models of the tunnel, bridge, and signal machine.

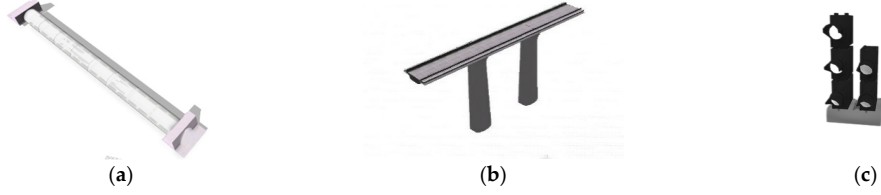

|     |     |     |
|:---:|:---:|:---:|
| (**a**) | (**b**) | (**c**) |

**Figure 7.** Three-dimensional symbol modelling: (**a**) bridge model; (**b**) tunnel model; (**c**) signal machine model.

③ Simplification of the three-dimensional symbol model. The three-dimensional symbols designed in this research had a multi-detail hierarchy. Therefore, when we browsed large-scale 3D scenes, the content information and colours should not be over-saturated. Priority should be given to expressing the geographical entity objects that the user focuses

on. We should simplify the process by combining the results of the classification while considering the overall 3D scene in the right level of detail. In small-scale scenes, the 3D symbols need to be simple, easy to read, and abstract while focusing on the more important objects. Therefore, we simplified the 3D symbols in five different layers of the model on the basis of the CityGML idea. This allows for more efficient loading and also provides more detailed content of the solid objects.

According to the above steps, we generated 3D symbols of the geographical elements in the region in different levels according to the five LODs. We imported all the 3D symbols into the database for storage.

Finally, we built the symbolic 3D high-speed railway scene platform. Different layers of 3D symbols were loaded at different scales by calling different 3D symbol services. One of the LOD0 contained important stations. As the scale level increased, geographical elements such as tunnels, bridges, roadbeds, and signalling machines appeared level by level. The multi-scale representation of the symbolic 3D high-speed railway scene was finally achieved. The experimental results are shown in Figure 8.

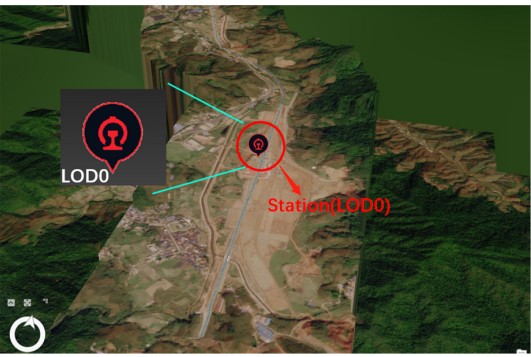

(**a**) Scene LOD0 with 3D symbol for station (LOD0)

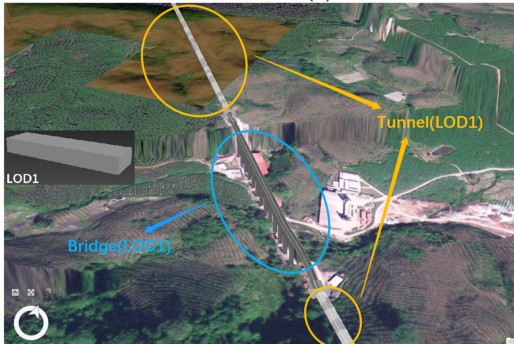

(**b**) LOD1 with 3D symbol for station (LOD1)

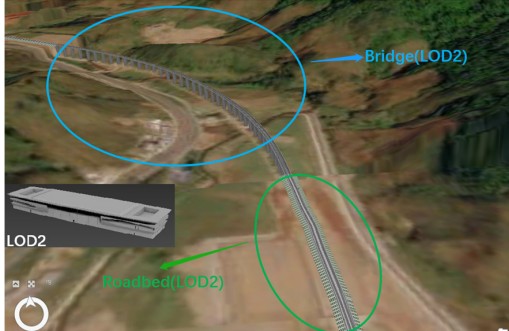

(**c**) LOD2 with 3D symbol for station (LOD2)

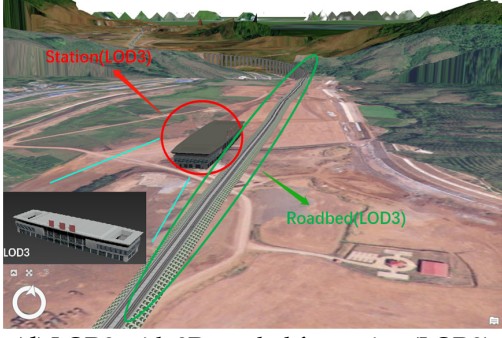

(**d**) LOD3 with 3D symbol for station (LOD3)

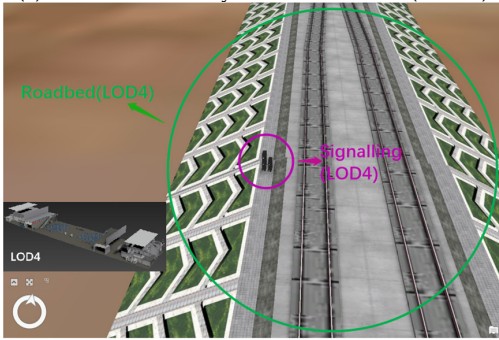

(**e**) LOD4 with 3D symbol for station (LOD4)

**Figure 8.** Multi-scale representation of symbolic 3D railway scenes: (**a**) LOD0: Only geographic entities with object importance level 5 were loaded, containing stations in the route and tunnels at

some junctions. The symbolic accuracy was at the lowest level of detail. (**b**) LOD1: Three-dimensional symbols of tunnels and some bridges are present in the scene. The accuracy of the 3D symbols at this level was improved compared to the previous level. (**c**) LOD2: In addition to the previously loaded stations, tunnels, and bridges, at this level of detail, some of the roadbeds out of junctions and turning points were loaded. The roadbeds were generally connected to the bridges. The 3D symbols at this level of detail had a clearer outline and colour. (**d**) LOD3: At this level, all the roadbeds are shown. The whole road was fully loaded, consisting of the tunnel, bridge, and roadbed stitched together. The 3D symbol structure at this scale was complete and well textured, especially the station model. (**e**) LOD4: At the highest level of detail scenes, all 3D symbols were at their highest accuracy. The 3D symbols of the signalling machines on the route were also loaded. These symbols were reproduced to the greatest extent possible for the real scene.

The real-time frame rate and scene loading speed were used as criteria for examination in the experimental results. The results of this research method were analysed in comparison with traditional 3D scene loading methods and are shown in Figure 9.

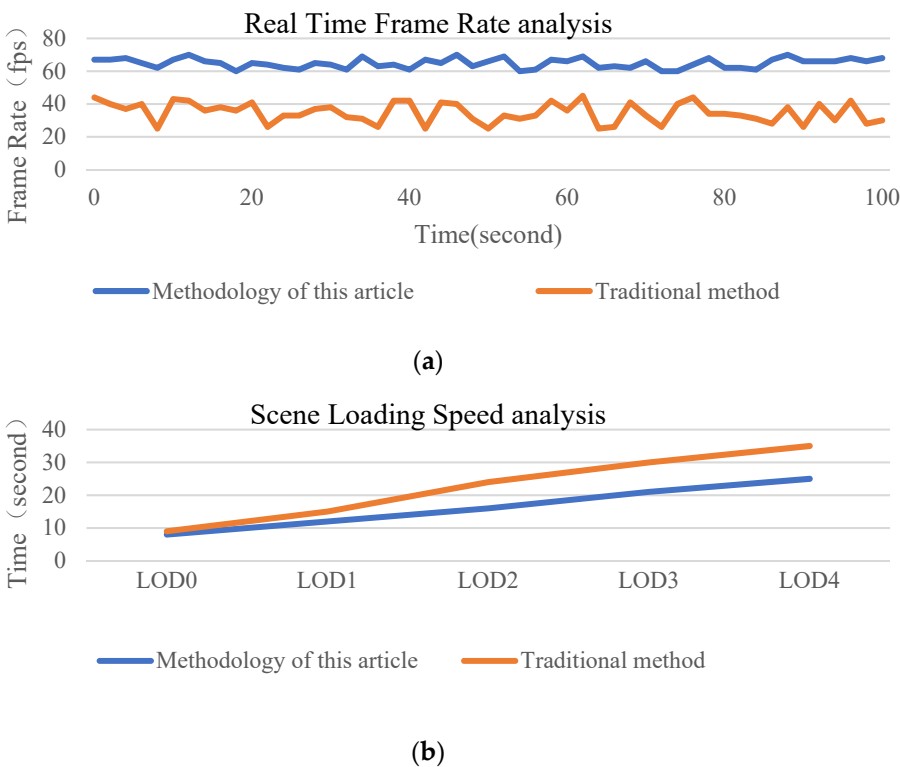

(**a**)

(**b**)

**Figure 9.** Quantitative comparative analysis: (**a**) The symbolic 3D railway scene data constructed by this research method had a higher frame rate after loading. Scene fluency was optimised. The user experience was improved. (**b**) Under the multi-scale constraint of this research method, the loading of geographic entities with focused features was faster. Moreover, the advantage became more obvious as the LOD level gradually increased. It improved the perceived depth of the user.

## 4. Conclusions

This research proposes a symbolic representation of 3D route scenes that takes into account the importance of objects. An object importance evaluation model was constructed by calculating the spatial importance of geographical elements in the 3D route scene and combining it with semantic importance. Moreover, according to the importance of each entity object in the 3D route scene, the corresponding 3D symbols were classified and symbolised, and a five-layer 3D route scene was constructed by drawing on the ideas of CityGML. This allowed the key elements in the 3D route scene to be highlighted and the scene to be loaded more effectively. Finally, the applicability of the method was verified using data from a railway as the basis for experiments. A five-level model of

the level of detail of the 3D railway scene from LOD0-LOD4 was created. This enabled the display of geographic entities with a high degree of importance at a large scale. An initial representation of their spatial location information as well as attribute information were available. Moreover, as the scale and viewing area was reduced, secondary objects were gradually shown, and the accuracy of the 3D symbols of important objects was improved. For important objects, the process of change from 2D symbols to geometry to textual features and internal structure occurred. This design makes it possible to fully represent important objects and to better meet the needs of the user. It also provides a more stable frame rate and faster loading of important objects in multi-scale 3D route scenes. It improves the user experience and perceived depth.

Symbolic 3D scenes take the user's perception of geospatial objects as a starting point. It is a new form of representation that highlights the object of focus. Combined with multi-scale expressions, it can better match the user's cognitive effect, especially for long-distance 3D scenes such as routes. The form of symbolic representation is more conducive to presenting information and aiding decision making. However, in the process of building 3D symbols, there is still much to be further investigated—for example, the design ideas of 3D symbols, and the mapping rules between attributes and 3D visual vectors. These are all valuable research directions that need to be studied in depth.

**Author Contributions:** Conceptualisation, L.H.; methodology, F.H.; software, F.H.; validation, T.S., X.Z., and T.Z.; formal analysis, X.Z. and N.M.; investigation, N.M.; resources, T.Z.; data curation, T.Z.; writing—original draft preparation, F.H.; writing—review and editing, L.H. and T.S.; visualisation, X.Z.; supervision, L.H.; project administration, X.Z. and T.Z.; funding acquisition, T.S. All authors have read and agreed to the published version of the manuscript.

**Funding:** This research was funded by the Research Project of China Academy of RAILWAY Sciences Corporation Limited (grant no. 2020YJ226). The authors wish to extend their sincere thanks for the support from China Academy of Railway Sciences.

**Institutional Review Board Statement:** Not applicable.

**Informed Consent Statement:** Not applicable.

**Data Availability Statement:** Not applicable.

**Acknowledgments:** The authors are thankful to Peng Bao for his contribution to the analysis and supervision of this article.

**Conflicts of Interest:** The authors declare no conflict of interest.

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
