# Peer review of "Research on the Symbolic 3D Route Scene Expression Method Based on the Importance of Objects"

_applsci, doi:10.3390/app122010532_

Round 1

Reviewer 1 Report

The paper emphasizes an approach combining the object importance of  geographical entities in the scene and the idea of CityGML on five layers to divide the symbolic 3D railway scene into five layer scales, including a five layer 3D railway scene detail hierarchy model from LOD0-LOD4. The paper subject is interesting, but it must to be highlighted better, especially the Conclusions section. 

line 29 - suggestion: "digital maps"/"digital representations" instead of "electronic maps"

line 265 - suggestion: "Symbolized 3D Railway Scene Construction" instead of "Symbolic 3D Railway Scene Construction"

Author Response

Dear Reviewer,

Kind regards,
Mr. Han

Reviewer 2 Report

Dear authors,

I find this topic really interesting and necessary. Anyway, I have to say that it is not easy to follow what is intended to say. This manuscript must be improved adding images and graphic examples for what is trying to be explained by words. Furthermore, the language is not very proper, being necessary to avoid the use of the second plural person, and expressions like "this paper".

Lines 73, 77 and 79 include many times the sentence "on the other hand".

At line 194 is refering to some image but it is not indicated which one.

At line 205 the writing is not serious.

The most important thing is that until "Results" there is no image of what the research is about. It is necessary to examplify the levels of details, clasify in a good manner the elements that can be found on a route, and make easier to follow this work. I thing is a very interesting work but must be deeply improved.

Author Response

Dear Reviewer,

Kind regards,
Mr. Han

Reviewer 3 Report

Despite the interesting topic, paper is similar to the technical report not a research paper. The paper has not the right structure in the current version. Figures and tables should have more meaningful captions. They should be cited in the text of the paper.

Abstract doesn’t cover the proposed methodology of the paper. Also, the experimental results are not in the abstract.

Introduction section is not similar to the academic research introductions. No enough literature review is performed.

Text should be revised and rewrite for English language correction. Even the title of the paper needs correction.

Most of the paper in the results section is linguistic description of the work, without any quantitative results and discussion.

Author Response

Dear Reviewer,

Kind regards,
Mr. Han

Round 2

Reviewer 1 Report

The authors managed to implement the suggestions. 

Author Response

Dear Reviewer

Sincerely Yours, Mr.Han

Reviewer 2 Report

Dear authors, 

Thanks for having into account the comments and improving this manuscript. It is necessary to pay attention to the language. In the title, it is said "base on" instead of "based on", and so it is said many times after. Furthermore, it is necessary to avoid the use of "we", "our", being necessary to write an impersonal text, as a scientific text, that is what you are doing.

The manuscript has improved with the introduction and the different figures that have been included. There is a gap between lines 413 and 414.

Good job.

Author Response

Dear Reviewer

Sincerely Yours, Mr.Han

Reviewer 3 Report

The progress in the structure of the revised paper is good. But, still some more corrections need:

Abstract doesn’t have any quantitative results. It is suggested to provide some quantitative results to clear the performance of your proposed methodology.

The revised introduction section has enough literature review. Please revise some mistakes in the citing references in the text (for instance page 2 paragraph 1, lines 47 and 50) and correct writing mistakes (for instance page 3, line 114 and 115). Please make more accurate revision on the whole of the text.

It is suggested to list the main contributions of your research in the end of the introduction section.

Results and graphical explanations are perfect in the revised version. Please make more discussions on the conclusion section.

Author Response

Dear Reviewer

Sincerely Yours, Mr.Han
